# Study on Creepage Control for PLS-160 Wheel–Rail Adhesion Test Rig Based on LADRC

**DOI:** 10.3390/s23041792

**Published:** 2023-02-05

**Authors:** Chun Tian, Gengwei Zhai, Yingqi Gao, Chao Chen, Jiajun Zhou

**Affiliations:** Institute of Rail Transit, Tongji University, Shanghai 201804, China

**Keywords:** PLS-160 wheel–rail adhesion test rig, nonlinear disturbance, linear active disturbance rejection controller, co-simulation, creepage control

## Abstract

Aiming at the problem of low control accuracy caused by nonlinear disturbances in the operation of the PLS-160 wheel–rail adhesion test rig, a linear active disturbance rejection controller (LADRC) suitable for the wheel–rail adhesion test rig was designed. The influence of nonlinear disturbances during the operation of the test rig on the control accuracy was analyzed based on SIMPACK. The SIMAT co-simulation platform was established to verify the control performance of the LADRC designed in this paper. The simulation results show that the speed and creepage errors of the test rig under the control of the LADRC met the adhesion test technical indicators under four different conditions. Compared with the traditional PID controller, the creepage overshoot and response time with the LADRC were reduced by 1.27% and 60%, respectively, under the constant creepage condition, and the stability recovery time was shorter under the condition of a sudden decrease in the adhesion coefficient. The LADRC designed in this paper shows better dynamic and anti-interference performance; therefore, it is more suitable for a follow-up study of the PLS-160 wheel–rail adhesion test rig.

## 1. Introduction

When a train runs on a rail surface with pollutants such as water and engine oil, low adhesion and large creep occur, which seriously affect traction/braking performance and operation safety. The existing wheel–rail adhesion theories mainly study the small creep stage without pollutants on the rail surface, which cannot truly reflect the wheel–rail state in the large creep stage with pollutants. The existing wheel–rail adhesion test rigs are mainly double-disc rolling test rigs [1,2,3,4], which cannot measure and quantitatively control the film thickness of rail surface pollutants. Therefore, the Braking Technology Institute of Tongji University designed a new type of circulator wheel–rail adhesion test rig, called PLS-160, which is used in the large creep condition and can truly and accurately reproduce low-adhesion states on actual track lines. Due to the particularity and uncertainty of the wheel–rail adhesion characteristic curve under the large creep condition, there are higher requirements for the stability and accuracy of the test rig speed and creepage control. There are various nonlinear disturbances during the operation of the test rig, which easily cause creepage to fluctuate, and it is difficult to obtain an accurate wheel–rail adhesion characteristic curve. Therefore, studying the creepage control method and controlling the speed and creepage error of the test rig within the allowable range are of great significance to reduce data errors and improve the test accuracy, which will not only be conducive to follow-up adhesion test research but also provides a certain reference value for the practical application of precise mechanical equipment control methods.

As a problem with great research value in the control field, disturbance rejection has attracted the attention of many researchers in recent years. Shen et al. proposed data-driven control strategies for networked nonlinear systems with event-triggered output [5]. The proposed strategies were verified by numerical examples. Gu et al. discussed event-triggered adaptive disturbance–observer-based control for Markov jump systems with general transition probabilities and multiple disturbances and proposed threshold-dependent anti-disturbance control schemes under different triggering rules to release the network bandwidth and reject disturbances [6]. They verified the validity of the proposed schemes using a numerical simulation comparison.

The PLS-160 wheel–rail adhesion test rig uses a permanent magnet synchronous motor (PMSM) as the power source, and its control methods are various. At present, many scholars have carried out relevant research. The control methods of PMSM mainly include proportional–integral–differential (PID) control, adaptive control, model predictive control (MPC), and active disturbance rejection control (ADRC). The classical PID control method does not depend on the model. The method is relatively mature and easy to implement, so it has a wide range of applications. However, the traditional PID controller has a long adjustment time and large overshoot, and its dynamic performance is not ideal. To address these problems, Cui et al. put forward an adaptive variable universal fuzzy PID compound control strategy and applied it to the vector control of the PMSM servo system based on the speed loop [7]. The strategy improves the response speed, anti-interference ability, and robustness of the system. Zeng et al. adopted fuzzy PID control for a gas turbine generator set and studied its dynamic response characteristics based on MATLAB/Simulink [8]. The results showed that the response speed of the generator set was improved and the overshoot was reduced. The adaptive control algorithm could compensate for the change in controlled object parameters to adapt to the change in controlled object characteristics. Yan et al. presented an improved sliding mode model reference adaptive system (SM-MRAS) speed observer for fuzzy control of a direct-drive wind power generation system with a permanent magnet synchronous generator (PMSG) [9]. Through a simulation and experiment, it was proved that the designed fuzzy controller had a faster dynamic response speed than the conventional PI controller. Lv et al. proposed a new fuzzy logic method for approximating nonlinear functions and disturbances in the system and designed an adaptive fuzzy tracking controller based on the backstepping method [10]. The research results showed that the method had good tracking performance. However, the adaptive control performance was affected by the controller adjustment speed, which could not guarantee that the system would always be stable in the parameter adjustment process. MPC adopts the rolling optimization strategy and can make up for the uncertainties caused by model mismatch, distortion, and disturbance in time, so as to obtain better dynamic control performance. Zhou et al. proposed a multi-vector finite control set model predictive control (MV-FCS-MPC) scheme based on fuzzy logic [11]. Through simulation and experiment, and compared with the traditional FCS-MPC method, they showed that this method had good dynamic and static performance and strong robustness. In [12], a novel model predictive direct torque control was proposed, and the authors proved through simulation that the method had good control performance. However, MPC requires high accuracy of the model and a large amount of calculation. There is a certain deviation between the system’s predicted output and its actual output. ADRC technology has strong adaptability to factors such as external environment interference and system parameter uncertainty. It has received more research attention in recent years. Zhang et al. proposed an improved variable structure active disturbance rejection control strategy, which greatly reduced the position tracking error and enhanced the anti-interference ability of the servo drive system [13]. Sun et al. improved the ADRC and applied the improved control algorithm to the 3.5 kW PMSM experimental platform [14]. The research results showed that the improved algorithm could ensure good dynamic and static performance of the motor. However, there are too many parameters to be tuned in ADRC and it is difficult to debug on-site. Nonlinear functions in ADRC lead to a complex structure of the controller and large amounts of calculation.

There are strong nonlinear disturbances in the operation of the PLS-160 wheel–rail adhesion test rig. In order to ensure the accuracy of the test results, it is necessary to improve the control performance and ensure the control accuracy. The parameters tuned by the PID controller cannot effectively adapt to interference factors in the system or the environment. The adaptive control and MPC methods largely depend on the model accuracy, but in practical application, it is difficult to obtain accurate system parameters of the test rig due to a variety of environmental interference factors. The ADRC controller has too many parameters to be tuned, so it is difficult to apply in practice. Therefore, there is presently no controller suitable for precise creepage control of the PLS-160 wheel–rail adhesion test rig.

To achieve accurate creepage control of the PLS-160 wheel–rail adhesion test rig, in this paper we analyzed the nonlinear disturbances existing in its actual operation. The SIMPACK dynamic simulation model of the test rig was established with the nonlinear disturbances as the input parameters. The dynamic response of the test rig under the motor open-loop control was analyzed. A linear active disturbance rejection controller (LADRC) suitable for the PLS-160 wheel–rail adhesion test rig was designed based on the double closed-loop speed and torque control strategy. The SIMAT co-simulation platform was established to verify the control performance and effect of the creepage controller designed in this paper.

## 2. PLS-160 Wheel–Rail Adhesion Test Rig

To study the wheel–rail adhesion characteristic of a train under large creep conditions, the Braking Technology Institute of Tongji University designed a new type of circulator wheel–rail adhesion test rig, called PLS-160, with a design speed of up to 160 km/h. Figure 1 shows the PLS-160 wheel–rail adhesion test rig.

The test rig is co-driven by the vehicle speed motor and wheel speed motor. One wheel is in a pure rolling state when running, and its speed simulates the vehicle speed *v*_c_. The other wheel is in the state of rolling with slip, and its speed simulates the axle speed $v_{w}$ of the train. Both wheels are equipped with speed sensors. According to the creepage calculation formula, $s=\left(v_{c}-v_{w}\right)/v_{c}$, the creepage data at different moments during the test can be measured. The axles of the test rig are equipped with torque sensors, which can measure the torque T at different moments. By inflating air into the cylinders above the wheels to simulate the axle load N, the wheel radius R is known, so the adhesion coefficient at different moments is μ=T/RN. According to the data of creepage and the adhesion coefficient, the adhesion coefficient–creepage curve can be drawn, and the wheel–rail adhesion characteristics can be studied.

The technical indicators of the adhesion test on the PLS-160 according to the need to find the second peak point in the large creep stage [15] are shown in Table 1. The speed error should be controlled within 1 km/h and the slip rate error should be controlled within 1%.

## 3. Nonlinear Disturbance Analysis of Wheel–Rail Adhesion Test Rig

There are various nonlinear disturbances during the operation of the PLS-160 wheel–rail adhesion test rig. To suppress the influence of nonlinear disturbances on creepage control and make the control more accurate, the main nonlinear disturbances during the test, including air resistance, adhesion coefficient–creepage characteristic, and mechanical structure transmission vibration, were analyzed.

### 3.1. Air Resistance

The adhesion test rig uses the rotation motion of the rotary arm to simulate the vehicle running speed. There is a nonlinear air resistance effect in the rotation process of the rotary arm, and its value is proportional to the speed quadratic. When the test rig is running at the maximum speed of 160 km/h, the rotation angular velocity of the rotary arm is up to 44.44 rad/s. The generated air resistance torque should be calculated through simulation to judge its impact on the operation of the test rig.

Based on CFD flow field simulation analysis, three-dimensional models of the adhesion test rig rotary arm and the flow field domain are established by using UG software, the model is meshed based on ANSYS ICEM, the flow field in the high-speed rotation of the adhesion test rig is simulated and numerically solved using Fluent, and the results are post-processed based on Tecplot. Figure 2 shows the CFD simulation calculation process.

The calculated air resistance torque of the rotary arm under different vehicle speed conditions through simulation is shown in Table 2.

According to Table 2, the power consumed by the rotation motion of the rotary arm under the extreme condition is P=Mω=154.44 N·m×44.44 rad/s≈6.9 kW, which is borne by the vehicle speed motor. The rated power of the vehicle speed motor is 79 kW, and the power consumed by the rotary arm can reach 8.7% of that rated power. Therefore, the air resistance during the adhesion test cannot be ignored, and it is a major nonlinear disturbance.

### 3.2. Adhesion Coefficient–Creepage Characteristic

Rail vehicle traction and braking are restricted by adhesion force, and the adhesion characteristic is a unique property of rail vehicles. The adhesion coefficient and the creepage between wheel and rail is a nonlinear relationship.

According to traditional adhesion theory, the adhesion coefficient increases linearly with creepage in the small creep area and presents zero slope or negative slope characteristics in the large creep area [16,17,18,19,20,21]. With increased creepage, the adhesion coefficient gradually increases to the limit value, and then remains the same or decreases continuously. However, more tests have shown that when there is a third-body medium between the wheel and the rail, the adhesion characteristic in the large creep area is inconsistent with traditional theory. The adhesion coefficient first decreases and then increases with increased creepage [22,23,24,25,26]. Figure 3 shows the nonlinear variation relationship between the adhesion coefficient and the creepage.

To sum up, the adhesion coefficient and creepage between the wheel and the rail show a nonlinear relationship of unknown size and variation trend in the large creep area, which leads to higher requirements for accurate adhesion test rig creepage control under the variable creepage condition. Therefore, the adhesion coefficient–creepage characteristic is also a nonlinear disturbance that cannot be ignored.

### 3.3. Mechanical Structure Transmission Vibration

The transmission system of the adhesion test rig is mainly composed of multiple bevel gear pairs, which will inevitably generate vibration during operation. As an elastic mechanical system, the gear transmission system will have the corresponding dynamic response with the influence of external dynamic excitation, and will generate nonlinear vibration, which makes it difficult to control the system accurately.

The external dynamic excitation of the gear transmission system can be categorized into two types: external and internal excitation. External excitation includes the excitation generated by the operation of the power sources (two drive motors of the adhesion test rig), the mechanical load, and other components such as rolling bearings in the mechanical transmission system. In particular, there is also the impact effect of the wheel and the rail in this system. The internal excitation of gears can be divided into four forms: stiffness, error, meshing impact, and gear tooth damage excitation. Stiffness excitation refers to the situation where the gears are engaged, and meshing stiffness and load show a periodic time-varying law, so a periodic time-varying gear force is applied to the system. Error excitation refers to machining and installation errors of the gears, causing a change in the instantaneous transmission ratio when engaging, resulting in periodic displacement excitation. Meshing impact excitation is caused by a gear approach point or recess point position error and elastic deformation caused by the gear on its own load when the gear enters the mesh. Gear tooth damage excitation refers to the impact excitation generated by gear tooth surface damage, such as peeling or root crack. Because internal excitation is related to the structural characteristics of the transmission system itself, whether the external excitation exists or not, the gear system will still be affected and will generate nonlinear vibrations.

To sum up, the excitation sources and types of vibration during the operation of the adhesion test rig are complex and diverse, and the generated vibration response is nonlinear and time-varying. Therefore, the influence of mechanical structure transmission vibration should be considered in the accurate control of the adhesion test rig.

As described in this section, the influence of nonlinear disturbances such as air resistance, adhesion coefficient–creepage characteristic, and mechanical structure transmission vibration cannot be ignored during the operation of the PLS-160 wheel–rail adhesion test rig, and should be considered in the creepage control of the test rig. As described in the following section, the creepage control of the adhesion test rig was studied by establishing simulation models.

## 4. Dynamic Simulation Model of Wheel–Rail Adhesion Test Rig

To verify the influence of nonlinear disturbances on the adhesion test rig and carry out creepage control on this basis, based on SIMPACK multi-body dynamic software, a dynamic model of the adhesion test rig was established, and the nonlinear disturbances were taken as the model input so that the model could fully reflect the dynamic characteristic of the rig during operation.

According to the actual bevel gear pair parameters of the test rig (see Table 3), the PLS-160 wheel–rail adhesion test rig model was established based on SIMPACK, including the rail, wheel, wheel axle, rotary arm, main shaft, shaft sleeve, bevel gear pair, and motor shaft. Each part was connected by force elements to simulate the interaction force that occurs during the operation of the test rig. The model consists of 16 bodies, 16 hinges, 13 force elements, and 16 degrees of freedom. Figure 4 shows the three-dimensional dynamic simulation model of the test rig and its topology.

The three nonlinear disturbances described in Section 3 were inputted into the simulation model. Mechanical structure transmission vibration was simulated by defining the gear contact force element, as shown in Figure 4b. The force element of air resistance was defined by the formula fitted by the data in Table 2 to simulate the air resistance torque, as shown in Figure 5a. An input function was defined by the example of the traditional nonlinear adhesion coefficient–creepage curve, which is relatively perfect for simulating the adhesion coefficient–creepage characteristic, as shown in Figure 5b.

The vehicle speed and wheel speed motor output shaft rotation speeds were set to simulate two creepage conditions (constant and variable creepage), and the dynamic response of the adhesion test rig under open-loop motor control was studied.

Under the constant creepage condition, the simulation vehicle speed was 50 km/h and the creepage was 0.1. The dynamic response of the adhesion test rig in this condition is shown in Figure 6.

It can be seen from Figure 6 that under the constant creepage condition, the actual wheel speed and creepage fluctuate around the target value. The maximum creepage error reaches 2.8%, which is more than 1%, so it does not meet the adhesion test technical indicators.

Under the variable creepage condition, the simulation vehicle speed was 50 km/h, the creepage changed in a sinusoidal function within the range of 0.01–0.3, and the change period was 6 s. The dynamic response of the adhesion test rig in this condition is shown in Figure 7.

It can be seen from Figure 7 that under the variable creepage condition, the actual wheel speed and creepage fluctuate around the target value, roughly showing a sine wave variation law. The maximum creepage error reaches 4.9%, which is much more than 1%, so it does not meet the adhesion test technical indicators.

As described in this section, the dynamic simulation model can reflect the influence on the dynamic characteristic of nonlinear disturbances of the test rig during operation. Under open-loop motor control, the maximum creepage error of the test rig reaches 4.9%, and does not meet the adhesion test technical indicators. Since the adhesion coefficient in the large creep area varies greatly with creepage, the small creepage control error will also lead to the amplification of the adhesion coefficient error in the tests, resulting in poor accuracy of the test results. Therefore, it is necessary to study the control strategy and the control algorithm to achieve accurate speed and creepage control of the test rig.

## 5. LADRC Based on Double Closed-Loop Speed and Torque Control Strategy

The control strategy and control algorithm comprise the creepage control core of the adhesion test rig. To achieve accurate creepage control, the basic framework of the creepage control strategy should be constructed first, and then the control algorithm should be studied and the controller designed on this basis.

### 5.1. Double Closed-Loop Speed and Torque Control Strategy

Since the power source of the PLS-160 wheel–rail adhesion test rig is the alternating current permanent magnet synchronous motor (ACPMSM), to achieve accurate creepage control, the control strategy framework can be divided into two parts: the ACPMSM control strategy and the creepage control strategy. Finally, based on the idea of cascade control, the creepage control strategy framework of the adhesion test rig is constructed.

The ACPMSM has a complex structure and is difficult to control. According to the characteristics of the PLS-160 wheel–rail adhesion test rig motor, a mathematical model of the ACPMSM is decoupled and its dimension is reduced based on the vector control method to make it equivalent to a DC motor for control. This paper uses the current feedback control method; under the control mode of $i_{d}=0$, the target torque current can be converted into the motor torque output. The specific formula is as follows:(1)$$T_{e}=\frac{3}{2}N_{p}\varphi_{f}i_{\mathrm{qref}}$$
where $T_{e}$ is the motor output torque, $N_{p}$ is the number of motor magnetic poles, $\varphi_{f}$ is the flux linkage peak value generated by the permanent magnet in stator phase winding, and $i_{\mathrm{qref}}$ is the target torque current.

The control of the adhesion test rig creepage is essentially the control of the simulated vehicle and axle speed. The specific formula is as follows:(2)$$s=\frac{v_{c}-v_{w}}{v_{c}}=1-\frac{R\omega_{1}}{\rho \omega_{2}}$$
where $v_{c}$ is the simulated vehicle speed, $v_{w}$ is the simulated axle speed, $\omega_{1}$ is the wheel axle rotation angular velocity, $\omega_{2}$ is the rotary arm rotation angular velocity, R is the wheel radius, and ρ is the rotary arm radius.

According to Formula (2), the creepage of the adhesion test rig is generated by wheel axle rotation angular velocity and rotary arm revolution angular velocity, and the essence of angular velocity control is motor output torque control. Therefore, the double closed-loop speed and torque control system is adopted as the framework, and the designed control strategy is shown in Figure 8. In the figure, the outer loop is speed control. The outer loop speed controller receives the error value between the actual vehicle or axle speed and the target speed and calculates the current $i_{\mathrm{qref}}$ required to control the simulated vehicle or axle speed. The inner loop is the torque control of the PMSM. According to Formula (1), the inner loop controller receives the error value between the actual and target current and corrects it to ensure that the motor output torque $T_{e}$ meets the outer loop speed control requirements. The inner loop torque control and outer loop speed control adjust mutually to ensure the accuracy of vehicle and axle speed, to ensure accurate creepage control of the adhesion test rig.

Based on the double closed-loop creepage control strategy framework in Figure 8, the creepage control algorithm was studied and the creepage controller was designed.

### 5.2. Study of LADRC for PLS-160 Wheel–Rail Adhesion Test Rig

At present, there are various control methods for the PMSM. In addition to the traditional PID control, the methods also include adaptive control, sliding film variable structure control, model predictive control, and active disturbance rejection control.

As mentioned in Section 5.1, the creepage control of the adhesion test rig is divided into two parts: speed and current (torque) control. For the current inner loop, because it has high sampling frequency, fast response speed, and less external interference, and because the process of extracting differential signals in PID control will amplify the noise, the proportional–integral (PI) control algorithm, with simple and easy-to-operate adjustment and higher reliability, is adopted. The sampling frequency of the speed outer loop is far less than that of the current inner loop, and there are various nonlinear disturbances in the system, so the linear active disturbance rejection control algorithm, with stronger robustness and convenient parameter adjustment, is adopted for controller design.

The basic structure of the current inner loop PI controller is as follows:(3)$$u_{q}=k_{\mathrm{pacr}}e_{i_{q}}\left(t\right)+k_{\mathrm{iacr}}\int e_{i_{q}}\left(t\right)dt$$
where $u_{q}$ is the output voltage of the PI controller, $k_{\mathrm{pacr}}$ and $k_{\mathrm{iacr}}$ are the proportional gain and integral gain, respectively, of the automatic current controller (ACR), and $e_{i_{q}}=i_{\mathrm{qref}}-i_{q}$ is the error between the target and the actual current.

Taking the wheel speed motor as an example, a speed outer loop LADRC suitable for the wheel–rail adhesion test rig was designed.

Based on the working principle of the wheel–rail adhesion test rig, the test rig stress analysis was carried out, as shown in Figure 9.

According to Figure 9, the dynamic equation of the adhesion test rig in the wheel axle rotation direction is as follows:(4)$$Fr_{1}+BR=J_{w}{\dot{\omega }}_{1}$$
where F is the force between the rotating output bevel gear pair, $r_{1}$ is the reference circle radius of the rotating output small bevel gear, B is the tangential force between wheel and rail, R is the wheel radius, and $J_{w}$ is the wheel axle rotation moment of inertia.

The torque equation of the rotation output small bevel gear is:(5)$$\frac{T_{\mathrm{we}}}{N_{w}}=Fr_{1}$$
where $T_{\mathrm{we}}$ is the wheel speed motor output torque and $N_{w}$ is the transmission ratio of the wheel speed motor transmission device.

If the target simulation wheel speed is set as $v_{\mathrm{wref}}$, Equation (4) becomes:(6)$$\frac{T_{\mathrm{we}}}{N_{w}}+BR=J_{w}\frac{\dot{v_{\mathrm{wref}}}}{R}$$

The motor vector control of Formula (1) is substituted into Formula (6), and the differential equation between the output current tuned by the current inner loop control and the simulated wheel speed is obtained:(7)$$J_{w}\frac{{\dot{v}}_{\mathrm{wref}}}{R}=\frac{3N_{p}\varphi_{\mathrm{fw}}}{2N_{w}}i_{\mathrm{qw}}+BR$$

If the control quantity of the current inner loop given by the speed outer loop is set as $i_{\mathrm{qw}}^{*}$, Formula (7) becomes:(8)$$J_{w}\frac{{\dot{v}}_{\mathrm{wref}}}{R}=\frac{3N_{p}\varphi_{\mathrm{fw}}}{2N_{w}}i_{\mathrm{qw}}^{*}+BR+\frac{3N_{p}\varphi_{\mathrm{fw}}}{2N_{w}}\left(i_{\mathrm{qw}}-i_{\mathrm{qw}}^{*}\right)$$

Based on the principle of active disturbance rejection control [27,28,29], the system balance equation becomes:(9)$${\dot{v}}_{\mathrm{wref}}=bi_{\mathrm{qw}}^{*}+\varphi \left(t\right)$$
where $b=\frac{3N_{p}\varphi_{\mathrm{fw}}R}{2N_{w}J_{w}}$ is the system input gain, $i_{\mathrm{qw}}^{*}$ is the input current of the controlled object, and φ(t) is regarded as the sum of the total disturbances composed of the internal and external disturbance of the system, with $\varphi \left(t\right)=\frac{BR^{2}}{J_{w}}+\frac{3N_{p}\varphi_{\mathrm{fw}}R}{2N_{w}J_{w}}\left(i_{\mathrm{qw}}-i_{\mathrm{qw}}^{*}\right)$.

The LADRC is mainly composed of linear extended state observer (LESO), linear state error feedback (LSEF), and other structures. LESO and LSEF are the results linearized by extended state observer (ESO) and nonlinear state error feedback (NLSEF), respectively, in ADRC [30].

According to the principle of LESO and Formula (9), the outer loop augmented form of the wheel speed motor after expansion is:(10)$$\{\begin{array}{c} {\dot{x}}_{1}=x_{2}+bu\\ {\dot{x}}_{2}=h\left(t\right)\\ y=x_{1} \end{array}$$
where $x_{1}=v_{\mathrm{wref}}$ and $x_{2}=\varphi \left(t\right)$ are the new state variables after expansion and ${\dot{x}}_{2}=h\left(t\right)$. If we make $X={\left[x_{1}\;x_{2}\right]}^{T}$, Equation (10) becomes the state space form:(11)$$\{\begin{array}{c} \dot{X}=AX+Bu+Hh\\ y=CX \end{array}$$
where $A=\left[\begin{array}{cc} 0 & 1\\ 0 & 0 \end{array}\right]$, $B=\left[\begin{array}{c} b\\ 0 \end{array}\right]$, $H=\left[\begin{array}{c} 0\\ 1 \end{array}\right]$, $C=\left[\begin{array}{cc} 1 & 0 \end{array}\right]$, and $u=i_{qw}^{*}$.

The differential equation of LESO is as follows:(12)$$\{\begin{array}{c} \varepsilon =x_{1}-y\\ {\dot{x}}_{1}=x_{2}-\beta_{01}\varepsilon +bu\\ {\dot{x}}_{2}=-\beta_{02}\varepsilon \end{array}$$

According to Formula (12), if the estimated value of $x_{1}$ by the observer is set as $z_{1}$, the estimated value of $x_{2}$ is $z_{2}$. If we make $\;Z={\left[z_{1}\;z_{2}\right]}^{T}$, the observer is constructed as follows:(13)$$\{\begin{array}{c} \dot{Z}=AZ+Bu-L(z_{1}-y)\\ z_{1}=CZ \end{array}$$
where $L=\left[\begin{array}{c} \beta_{01}\\ \beta_{02} \end{array}\right]$, and $\beta_{01}$ and $\beta_{02}$ are the correction gains of the observer.

According to the principle of LSEF, the control quantity $u_{0}\left(t\right)$ is designed as:(14)$$u_{0}=\beta_{1}\left(v_{0}-z_{1}\right)$$

By compensating the estimated disturbance value obtained by the observer to the control quantity, the final control quantity input form of the controlled object is:(15)$$u=u_{0}-\frac{z_{2}}{b}$$

After the above operations, the LADRC converts the nonlinear control system of the adhesion test rig into an integrated linear series system. To suppress the overshoot problem, a linear tracking differentiator (LTD) is added to the controller. Its structure is as follows:(16)$$\{\begin{array}{c} e=v_{\mathrm{wref}}-v_{0}\\ {\dot{v}}_{0}=-rfal\left(e,\alpha ,h_{0}\right) \end{array}$$
where the structural form of function fal is as follows:(17)$$fal\left(\varepsilon ,\alpha ,\delta \right)=\{\begin{array}{c} {\left|\varepsilon \right|}^{\alpha }\cdot sign\left(\varepsilon \right),\left|\varepsilon \right|>\delta \\ \frac{\varepsilon }{\delta^{\alpha -1}},\left|\varepsilon \right|\leq \delta \end{array}$$

To sum up, the structural diagram of the LADRC of the wheel speed motor is shown in Figure 10.

The stability of the LESO is analyzed as follows.

The observer error can be obtained by subtracting the system state space form (11) from the LESO Equation (13). Let e=Z−X, then
(18)$$\dot{e}=\left(A-LC\right)e-Hh$$
where e is the error of the observer.

The stability proof of the observer can be converted to the stability proof of Equation (18). The characteristic polynomial of the error Equation (18) is
(19)$$\lambda \left(s\right)=\left|sE-\left(A-LC\right)\right|=s^{2}+\beta_{01}s+\beta_{02}$$

According to the sufficient and necessary condition for the stability of the control system, if the system is stable, the roots of the characteristic equation must be located on the left half of the s plane, that is, s<0. According to the parameter configuration method based on bandwidth [30], we set all poles of the observer at $-\omega_{0}$, then
(20)$$s^{2}+\beta_{01}s+\beta_{02}={\left(\lambda +\omega_{0}\right)}^{2}$$
where $\omega_{0}$ is the bandwidth of the observer.

Therefore, the correction gains of the observer are
(21)$$\{\begin{array}{c} \beta_{01}=2\omega_{0}\\ \beta_{02}=\omega_{0}^{2} \end{array}$$

Setting $\beta_{01}$ and $\beta_{02}$ in this way not only simplifies the observer correction gains to the adjustment of $\omega_{0}$, which greatly simplifies the parameter tuning process, but also ensures the stability of the system and the asymptotic convergence of the error Equation (18) [31]. Therefore, the system after parameter tuning is stable and the outputs $z_{1}$ and $z_{2}$ of the LESO can well estimate system state parameters $x_{1}$ and $x_{2}$.

## 6. Creepage Control Based on SIMAT Co-Simulation Platform

### 6.1. Establishment of SIMAT Co-Simulation Platform

To verify the creepage control performance of the LADRC, described in Section 5.2, based on the dynamic simulation model in Section 4, the creepage controller model of the adhesion test rig was established based on MATLAB/Simulink, and the SIMAT co-simulation platform was established, as shown in Figure 11. This platform can realize a real-time data exchange between the creepage control part and the dynamic simulation part.

The SIMAT co-simulation platform is mainly composed of five parts: creepage calculation module, starting module, speed outer loop controller, PMSM torque controller, and dynamic simulation module, respectively, corresponding to Parts 1–5 in Figure 11. The creepage calculation module mainly converts the input target creepage into the target simulated vehicle and axle speeds. The starting module is used to switch the wheel speed motor output mode. The speed outer loop controller and the PMSM torque controller jointly constitute the LADRC based on the double closed-loop speed and torque control strategy described in Section 5. The dynamic simulation module is the SIMPACK dynamic model interface of the adhesion test rig. It is responsible for real-time data transmission and the exchange of the MATLAB/Simulink and SIMPACK simulation programs.

Table 4 shows the main parameters of the co-simulation platform.

### 6.2. Simulation Results and Discussion

To verify the control effect and adaptability to different conditions of the creepage controller designed in this paper, four simulation conditions were set up: constant creepage, variable creepage, sudden decrease in adhesion coefficient, and variable adhesion characteristic. Based on the SIMAT co-simulation platform, the control effects of the traditional PID controller and LADRC designed in this paper were compared and analyzed.

#### 6.2.1. Constant Creepage Condition

Under the constant creepage condition, the target simulation vehicle speed was set at 50 km/h, with 0–1 s as the test rig starting process, and the target creepage was 0.1 after 1 s. The dynamic response of the adhesion test rig simulation model under the control of the PID controller and the LADRC is shown in Figure 12.

It can be seen from Figure 12 that the wheel speed error under PID and LADRC control in the condition of constant creepage is less than 1 km/h, and the creepage error is less than 1%, both of which meet the adhesion test technical indicators. Table 5 shows the creepage control performance indicators of the two controllers.

According to Table 5, compared with the PID controller, the overshoot of the LADRC designed in this paper is reduced by 1.27%, the response time is shortened by 60%, the response speed is greatly improved, and the dynamic performance is better.

#### 6.2.2. Variable Creepage Condition

Under the variable creepage condition, the target simulation vehicle speed was set at 50 km/h, with 0–1 s as the test rig starting process, creepage changed in a sinusoidal function within the range of 0.01–0.3 after 1 s, and there was a change period of 6 s. The dynamic response of the adhesion test rig simulation model under the control of the PID controller and the LADRC is shown in Figure 13.

It can be seen from Figure 13 that after the test rig starts, the actual wheel speed and creepage change periodically with time, showing a good following performance. The wheel speed and creepage errors under PID and LADRC control both meet the adhesion test technical indicators.

#### 6.2.3. Condition of Sudden Decrease in Adhesion Coefficient

Under the condition of a sudden decrease in the adhesion coefficient, the target simulation vehicle speed was set at 50 km/h, with 0–1 s as the test rig starting process, target creepage of 0.1 after 1 s, and a sudden decrease in the adhesion coefficient to 1/5 of the original after 2 s to simulate wheels running on a clean and smooth rail surface to a low-adhesion rail surface with pollutants. The dynamic response of the adhesion test rig simulation model under the control of the PID controller and the LADRC is shown in Figure 14.

It can be seen from Figure 14 that when the adhesion coefficient suddenly decreases at 2 s, the speed and creepage curves under the regulation of the LADRC and PID controllers return to the stable state after a short fluctuation, indicating that both controllers have certain anti-interference performance. However, it can be seen that LADRC takes a shorter time to restore stability, is impacted less by the sudden decrease in adhesion coefficient, and has stronger anti-interference performance.

#### 6.2.4. Variable Adhesion Characteristic Condition

According to Section 3.2, when there is a third-body medium between the wheel and the rail under the large creep condition, the adhesion coefficient will first decrease and then increase with increased creepage, which is inconsistent with the traditional adhesion theory. Therefore, based on the measured data of a test rig, the adhesion coefficient–creepage characteristic curve was redefined in SIMPACK under the variable adhesion characteristic condition, as shown in Figure 15.

The target simulation vehicle speed was set at 50 km/h, with 0–1 s as the test rig starting process, the creepage changed in a sinusoidal function within the range of 0.01–0.3 after 1 s, and the change period was 6 s. The dynamic response of the adhesion test rig simulation model under the control of the PID controller and the LADRC is shown in Figure 16.

It can be seen from Figure 16 that the wheel speed and creepage under PID and LADRC control both show good following performance, and the wheel speed and creepage errors are both within the error control range, meeting the adhesion test technical indicators.

In summary, under conditions of constant creepage, variable creepage, sudden decrease in adhesion coefficient, and variable adhesion characteristic, the speed error under LADRC control designed in this paper is less than 1 km/h and the creepage error is less than 1%, meeting the adhesion test technical indicators. Compared with the traditional PID controller, the LADRC designed in this paper has faster response speed, less overshoot, stronger anti-interference performance, and shows good adaptability to different nonlinear adhesion characteristic curves. Therefore, the LADRC designed in this paper can ensure the accuracy of adhesion test results and is more suitable for the follow-up study of the PLS-160 wheel–rail adhesion test rig.

At present, the PLS-160 wheel–rail adhesion test rig has been put into use, and the creepage errors under both constant and variable creepage conditions meet the technical indicators, showing good control stability and following performance, as shown in Figure 17.

In the future, a series of adhesion tests will be carried out based on the PLS-160 wheel–rail adhesion test rig under the large creep condition with different third-body media between the wheel and the rail.

## 7. Conclusions

In this paper, an LADRC suitable for the PLS-160 wheel–rail adhesion test rig was designed and its control effect was verified by co-simulation.

(1) The SIMPACK dynamic simulation model of the PLS-160 wheel–rail adhesion test rig was established, various nonlinear disturbances existing in its operation process were inputted, and the test rig model was simulated. The results show that the creepage error of the test rig under open-loop motor control was up to 4.9%, which does not meet the adhesion test technical indicators. It is necessary to accurately control the creepage of the test rig.

(2) Based on the double closed-loop speed and torque creepage control strategy, an LADRC suitable for the PLS-160 wheel–rail adhesion test rig was designed. The SIMAT co-simulation platform was established to study the control performance of the LADRC designed in this paper. The results show that under various conditions, the speed error of the adhesion test rig under LADRC control was less than 1 km/h and the creepage error was less than 1%, meeting the adhesion test technical indicators. The LADRC designed in this paper can adapt to a variety of complex conditions, showing good control stability and following performance.

(3) Compared with the traditional PID controller, the creepage overshoot of the adhesion test rig under the LADRC control designed in this paper was reduced by 1.27%, the response time was shortened by 60% under the constant creepage condition, and the stability recovery time was shorter under the condition of a sudden decrease in adhesion coefficient, showing better dynamic and anti-interference performance. Therefore, this LADRC is more suitable for follow-up study of the PLS-160 wheel–rail adhesion test rig.

(4) In addition to accurately controlling creepage for the PLS-160 wheel–rail adhesion test rig, the principle of LADRC designed in this paper is also applicable to other mechanical equipment that is subjected to large nonlinear disturbances and needs to be accurately controlled. In the future, the parameters of the controller will be further adjusted and optimized to give the control system better robustness and anti-interference ability, and to improve adhesion test accuracy.

## Figures and Tables

**Figure 1 sensors-23-01792-f001:**
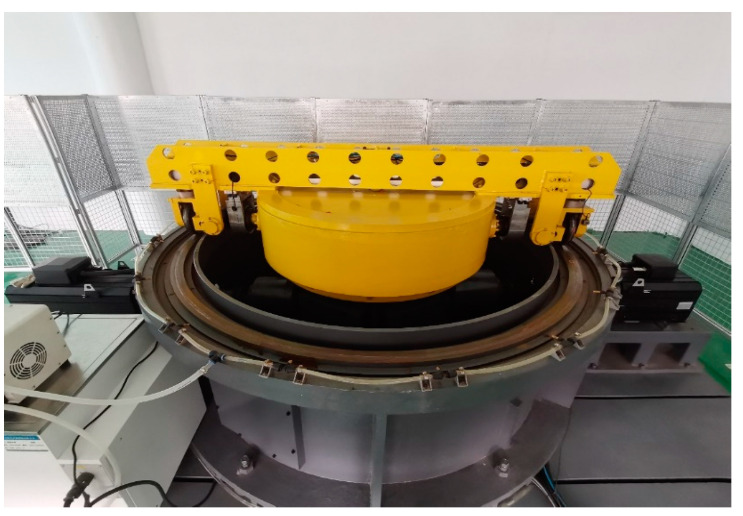
PLS-160 wheel–rail adhesion test rig.

**Figure 2 sensors-23-01792-f002:**
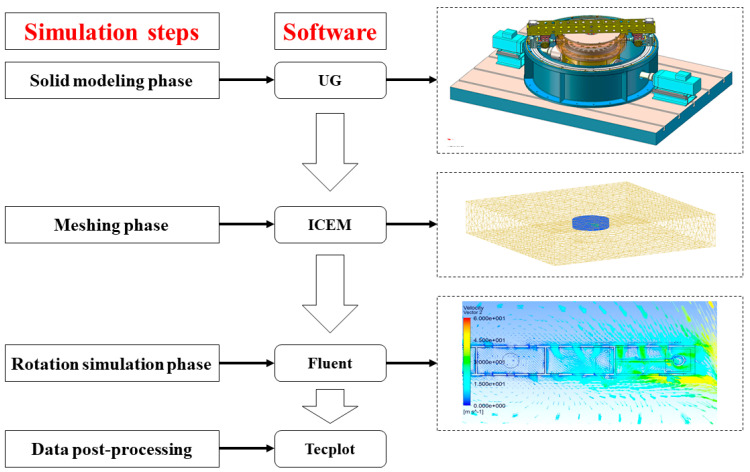
Diagram of CFD flow field simulation calculation process.

**Figure 3 sensors-23-01792-f003:**
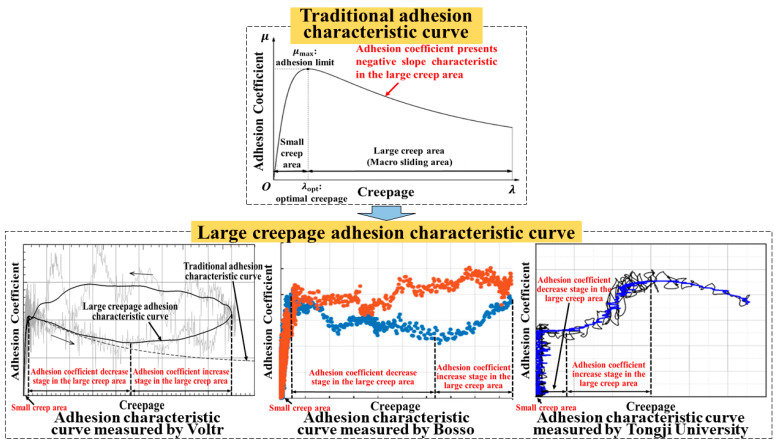
Diagram of nonlinear variation relationship between adhesion coefficient and creepage [20,21,22].

**Figure 4 sensors-23-01792-f004:**
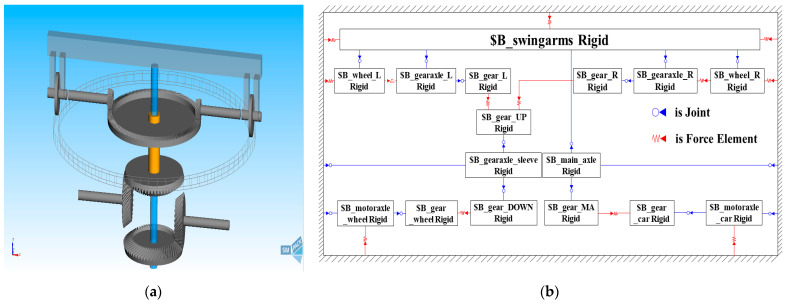
(**a**) Three-dimensional dynamic simulation model of wheel–rail adhesion test rig; (**b**) topology diagram of test rig model.

**Figure 5 sensors-23-01792-f005:**
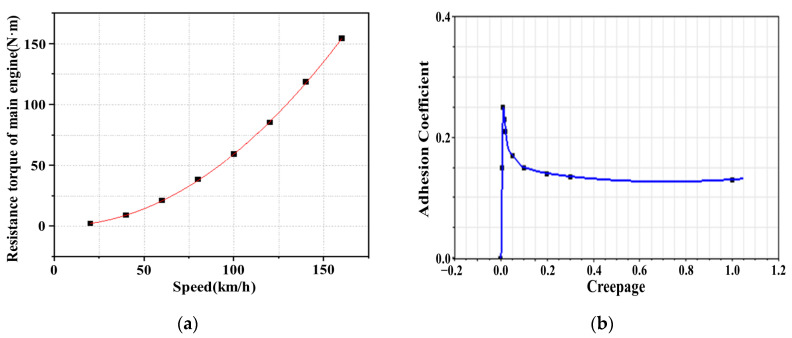
Diagram of nonlinear disturbance input for test rig model: (**a**) air resistance torque at different simulation vehicle speeds; (**b**) wheel–rail adhesion coefficient–creepage characteristic.

**Figure 6 sensors-23-01792-f006:**
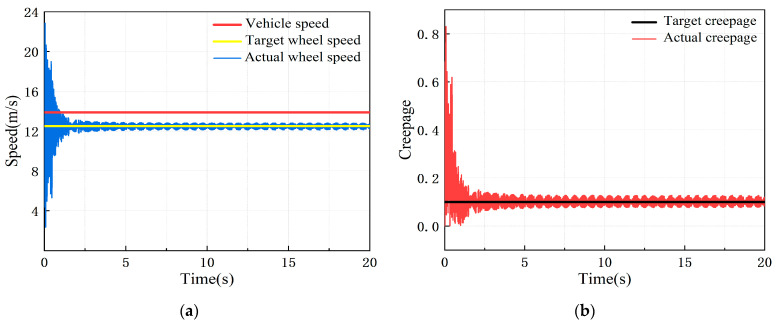
Dynamic response of test rig simulation model under constant creepage condition: (**a**) speed response curve; (**b**) creepage response curve.

**Figure 7 sensors-23-01792-f007:**
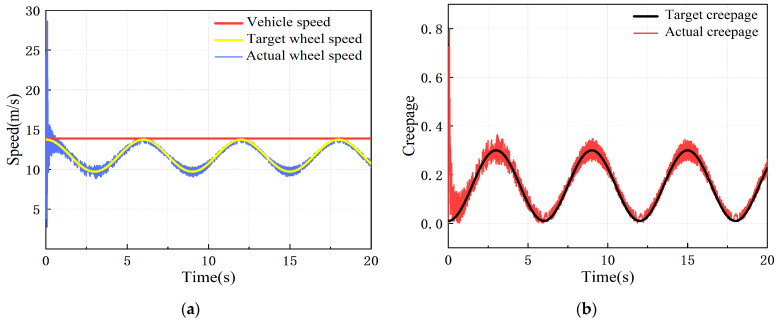
Dynamic response of test rig simulation model under variable creepage condition: (**a**) speed response curve; (**b**) creepage response curve.

**Figure 8 sensors-23-01792-f008:**
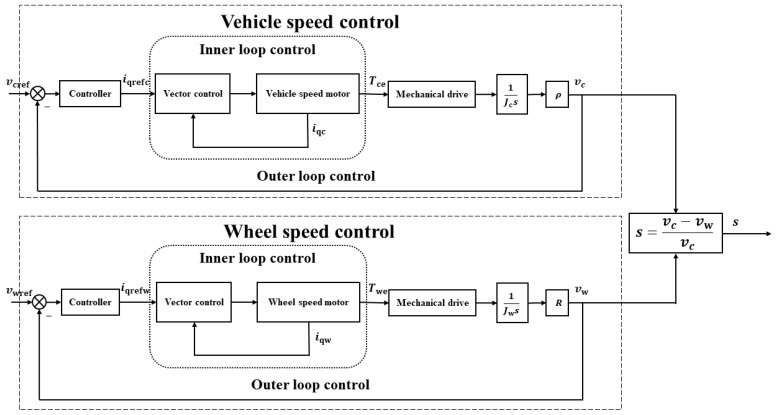
Diagram of double closed-loop creepage control strategy framework.

**Figure 9 sensors-23-01792-f009:**
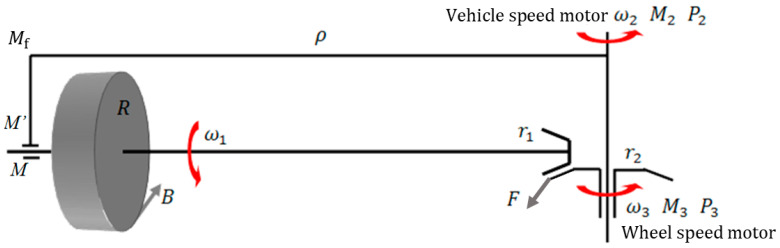
Diagram of test rig stress analysis.

**Figure 10 sensors-23-01792-f010:**
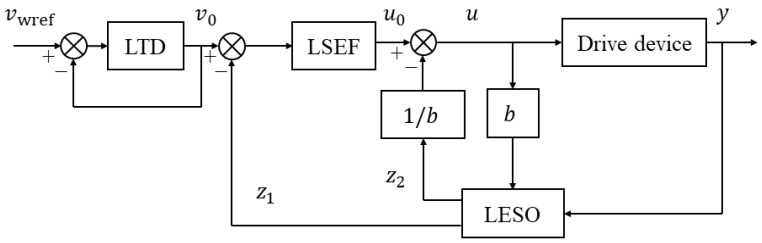
Structural diagram of LADRC of speed outer loop.

**Figure 11 sensors-23-01792-f011:**
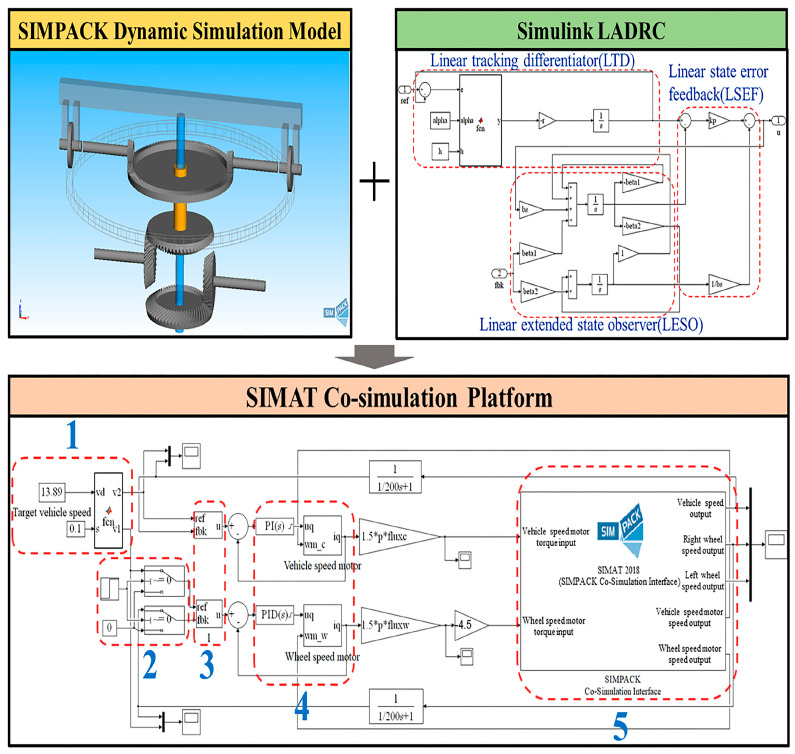
SIMAT co-simulation platform.

**Figure 12 sensors-23-01792-f012:**
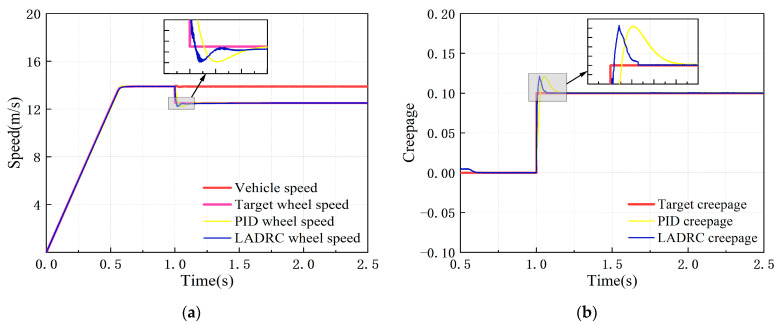
Dynamic response of test rig simulation model after controlling under constant creepage condition: (**a**) speed response curve; (**b**) creepage response curve.

**Figure 13 sensors-23-01792-f013:**
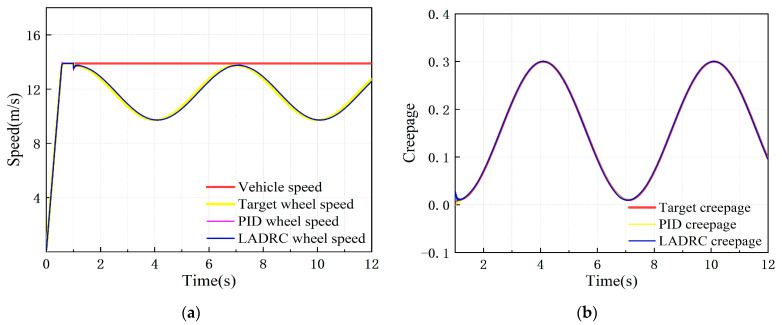
Dynamic response of test rig simulation model after controlling under variable creepage condition: (**a**) speed response curve; (**b**) creepage response curve.

**Figure 14 sensors-23-01792-f014:**
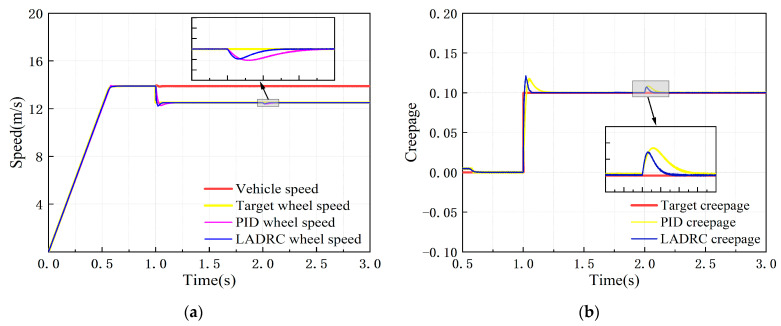
Dynamic response of test rig simulation model after controlling under sudden decrease in adhesion coefficient condition: (**a**) speed response curve; (**b**) creepage response curve.

**Figure 15 sensors-23-01792-f015:**
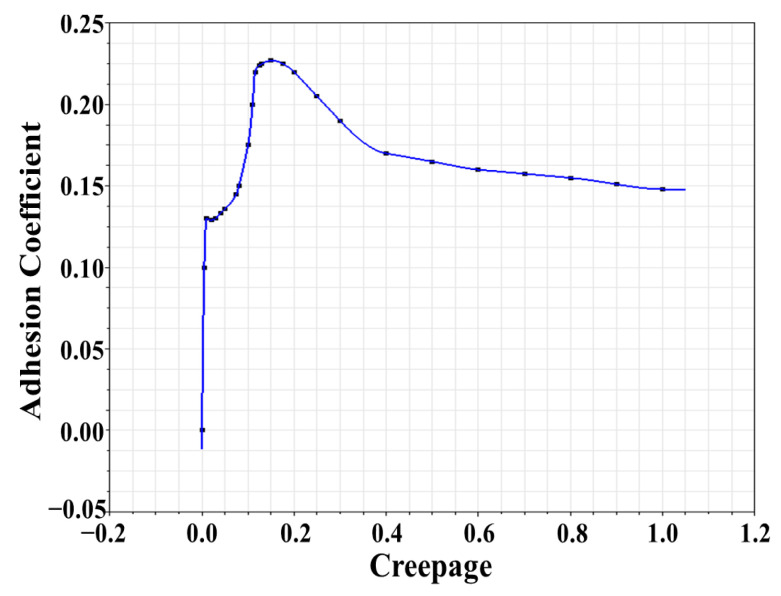
Wheel–rail adhesion characteristic curve under large creepage condition.

**Figure 16 sensors-23-01792-f016:**
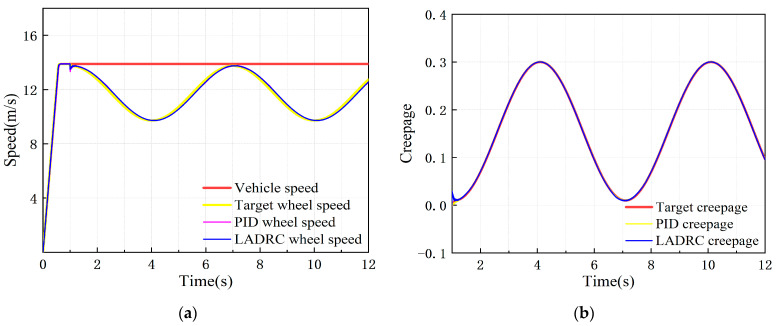
Dynamic response of test rig simulation model after controlling under variable adhesion characteristic condition: (**a**) speed response curve; (**b**) creepage response curve.

**Figure 17 sensors-23-01792-f017:**
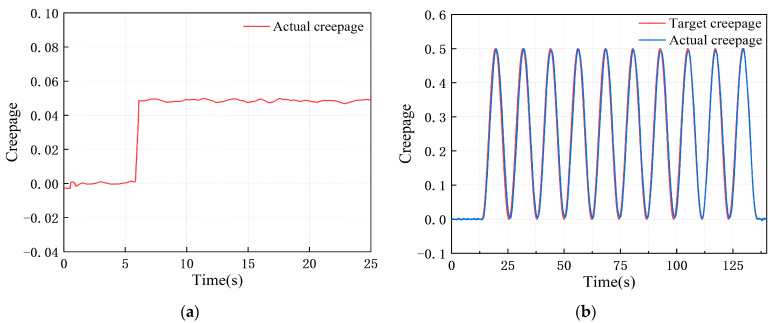
Dynamic response of test rig under different creepage conditions: (**a**) constant creepage; (**b**) variable creepage.

**Table 1 sensors-23-01792-t001:** Adhesion test technical indicators.

Error Range	Indicator
Speed control error range	<1 km/h
Creepage control error range	<1%

**Table 2 sensors-23-01792-t002:** Air resistance torque under different vehicle speed conditions.

**Vehicle speed (km/h)**	20	40	60	80	100	120	140	160
**Air resistance torque (N·m)**	2.38	8.99	21.06	38.5	59.29	85.45	118.67	154.44

**Table 3 sensors-23-01792-t003:** Bevel gear pair parameters of test rig.

Gear Type	Number of Teeth	Module (mm)	Reference Cone Angle (°)	Radial Modification Coefficient	Gear Ratio
Revolution input large gear	59	9	67.036	−0.141	2.36
Revolution input small gear	25	9	22.964	0.141	
Rotation input large gear	59	9	67.036	−0.141	2.36
Rotation input small gear	25	9	22.964	0.141	
Rotation output large gear	181	5	84.007	−0.228	9.53
Rotation output small gear	19	5	5.993	0.228	

**Table 4 sensors-23-01792-t004:** Parameters of SIMAT co-simulation platform.

Motor Type	Parameter	Value
Vehicle speed motor	Vehicle speed motor transmission ratio $N_{c}$	0.4237
	Vehicle speed motor winding resistance $R_{c}\;\left(\Omega \right)$	0.0295
	Vehicle speed motor winding inductance $L_{c}\;\left(H\right)$	1.1 × 10^−4^
	Vehicle speed motor maximum current $i_{c\_\max}\left(A\right)$	375
	Vehicle speed motor magnetic pole number $N_{\mathrm{pc}}$	4
	Vehicle speed motor equivalent moment of inertia at load end $J_{c}\;\left(\mathrm{kg}\cdot m^{2}\right)$	81.9942
	Vehicle speed motor voltage constant $K_{\mathrm{ec}}\;\left(V/1000\;\mathrm{rpm}\right)$	314
	Vehicle speed motor torque constant $K_{\mathrm{tc}}\;\left(N\cdot m/A\right)$	2.60
	Vehicle speed motor permanent magnet flux linkage $\varphi_{\mathrm{fc}\;}\left(\mathrm{Wb}\right)$	0.4327
Wheel speed motor	Wheel speed motor transmission ratio $N_{w}$	0.897
	Wheel speed motor winding resistance $R_{w}\;\left(\Omega \right)$	0.068
	Wheel speed motor winding inductance $L_{w}\;\left(H\right)$	2 × 10^−4^
	Wheel speed motor maximum current $i_{w\_\max}\left(A\right)$	186
	Wheel speed motor magnetic pole number $N_{\mathrm{pw}}$	4
	Wheel speed motor equivalent moment of inertia at load end $J_{w}\;\left(\mathrm{kg}\cdot m^{2}\right)$	10.8449
	Wheel speed motor voltage constant $K_{\mathrm{ew}\;}\left(V/1000\mathrm{rpm}\right)$	317
	Wheel speed motor torque constant $K_{\mathrm{tw}}\;\left(N\cdot m/A\right)$	2.62
	Wheel speed motor permanent magnet flux linkage $\varphi_{\mathrm{fw}}\;\left(\mathrm{Wb}\right)$	0.4366

**Table 5 sensors-23-01792-t005:** Creepage control performance indicators of PID and LADRC.

Controller Type	Overshoot (%)	Response Time (s)
PID	21.13	0.15
LADRC	19.86	0.06

## Data Availability

Not applicable.

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
