# Peer review of "Study on Creepage Control for PLS-160 Wheel–Rail Adhesion Test Rig Based on LADRC"

_sensors, 2023, doi:10.3390/s23041792_

Round 1

Reviewer 1 Report

This paper is devoted to the LADRC based creepage control for PLS-160 wheel-rail adhesion 2 test rig. I recommend it to publish on Sensors once the following comments are well concerned.

1. The introduction should be updated with recent results to highlight the contributions of this work.

2. It is recommended to add some remarks below the main results to show technical difficulties or differences with existing results.

3. The paper is missing the necessary theoretical proof.

4. In Equation (14), the  signal $\upsilon_werf$ before filtering is used. However, as can be seen in Figure 10, the control quantity $u_0$ is a combination of $\upsilon_0$ and $z_1$. Please give some comments.

5. Please check the whole paper carefully to correct typos and syntax errors and make the presentation more fluent.

Reviewer 2 Report

Figure 3 is not readable.

Figure 4(b) is not readable.

Block diagram in Figure 11 is not readable.

Figure 15 is not readable.

Reviewer 3 Report

The paper presents a novel controller dedicated to a test rig for wheel-rail adhesion analysis, named PLS-160, available at the Braking Technology Institute of Tongji University.

The technical contents of the paper are nice and adequately conveyed.

The authors should however better highlight the value of their work by discussing how other researchers can take benefit of it. In other terms, they should underline in which way their paper is much more than a good technical report, which is the opinion that several readers could get if they read the paper in the present form. For instance they could insert some paragraphs in Introduction as well as in Conclusions to discuss how their proposal and the underneath ideas can be considered general achievements and not just the technical advancements of an equipment of their own Institute.

Round 2

Reviewer 1 Report

This paper is devoted to creepage Control for PLS-160 wheel-rail adhesion 3 test rig based on LADRC. Some comments are given below:

1. As it is known, disturbance rejection is an interesting problem in control field, therefore, it is suggested to  include some recent results on this aspect to highlight your contribution in Introduction, such as  Extended Disturbance-Observer-Based Data-Driven Control of Networked Nonlinear Systems with Event-Triggered Output,Event-triggered control of Markov jump systems against general transition probabilities and multiple disturbances via adaptive-disturbance-observer approach.

2. The authors claim 'a new linear active disturbance rejection controller'. Please show this aspect more clearly, especially to compare with existing results.

3. Since a new linear active disturbance rejection controller is proposed, it is suggested to do the stability analysis of the resultant closed-loop systems as Disturbance-Observer-Based Data-Driven Control of Networked Nonlinear Systems with Event-Triggered Output,Event-triggered control of Markov jump systems against general transition probabilities and multiple disturbances via adaptive-disturbance-observer approach.

4. There are some typos and syntax errors, please check the whole paper carefully to correct them and polish the language.
